# Assessment of Heavy Metal Accumulation in Wastewater–Receiving Soil–Exotic and Indigenous Vegetable Systems and Its Potential Health Risks: A Case Study from Blantyre, Malawi

**DOI:** 10.3390/ijerph22111614

**Published:** 2025-10-23

**Authors:** Chimwemwe Chiutula, Andrew G. Mtewa, Amon Abraham, Richard Lizwe Steven Mvula, Alfred Maluwa, Fasil Ejigu Eregno, John Njalam’mano

**Affiliations:** 1Department of Energy Resources, Ndata School of Climate and Earth Sciences, Malawi University of Science and Technology, Limbe, Blantyre P.O. Box 5196, Malawi; moh-015-22@must.ac.mw; 2Department of Applied Studies, Chemistry Section, Malawi Institute of Technology, Malawi University of Science and Technology, Limbe, Blantyre P.O. Box 5196, Malawi; amtewa@must.ac.mw; 3Department of Earth Sciences, Ndata School of Climate and Earth Sciences, Malawi University of Science and Technology, Limbe, Blantyre P.O. Box 5196, Malawi; poh-004-22@must.ac.mw; 4Directorate of Research and Outreach, Malawi University of Science and Technology, Limbe, Blantyre P.O. Box 2109, Malawi; aomaluwa@must.ac.mw; 5Department of Building, Energy and Material Technology, Faculty of Engineering Science and Technology, UiT, The Arctic University of Norway, Postboks 385, 8514 Narvik, Norway; fasil.e.eregno@uit.no; 6Department of Water Resources Management, Ndata School of Climate and Earth Sciences, Malawi University of Science and Technology, Limbe, Blantyre P.O. Box 5196, Malawi; jnjalammano@must.ac.mw

**Keywords:** consumption, daily intake, health risks, heavy metals, vegetables, wastewater

## Abstract

Urban and peri-urban farmers in Malawi increasingly use treated and untreated wastewater for vegetable production, but little is known about the extent of heavy metal accumulation in both exotic and indigenous vegetables, particularly with respect to differences between edible tissues (leaves vs. stems). This study addresses this gap by measuring the concentrations of cadmium (Cd), chromium (Cr), lead (Pb), zinc (Zn), and copper (Cu) in wastewater, soils, and six vegetables including three exotic and three indigenous irrigated with effluent from the Soche Wastewater Treatment Plant in Blantyre. Metal concentrations were determined using Atomic Absorption Spectrophotometry. Wastewater contained Zn (0.01 ± 0.001 mg/L) and Cu (0.02 ± 0.018 mg/L), both below World Health Organization (WHO) and Malawi Bureau of Standards (MBS) limits (Zn: 0.2 mg/L; Cu: 2 mg/L), while Cd, Cr, and Pb were below detection limit. In soils, Zn reached 56.4 ± 0.5 mg/kg, exceeding the WHO limit of 36 mg/kg; other metals remained within WHO permissible values. Vegetables showed species- and tissue-specific variation in metal accumulation: Cr reached 4.65 mg/kg in *Cucurbita moschata* stems, Cd up to 0.31 mg/kg in *Amaranthus retro-flexus* leaves, and Pb up to 4.09 mg/kg in *Brassica rapa* stems—all above FAO/WHO permissible limits (2.3, 0.2, and 0.3 mg/kg, respectively). Duncan’s post hoc analysis confirmed significant differences (*p* < 0.05) across matrices and plant parts, with leaves generally accumulating more Zn and Cu than stems. Principal component analysis (PCA) revealed that Zn, Cu, Cr, and Pb in the wastewater-soil-vegetable system largely share a common source, likely wastewater effluent and historical soil contamination, while Cd showed a more sporadic distribution, highlighting differential accumulation pathways. Health risk assessments revealed high Health Risk Index (HRI) values, with *Brassica rapa* stems (HRI = 92.3) and *Brassica rapa* subsp. *chinensis* leaves (HRI = 82.2) exceeding the safe threshold (HRI > 1), indicating potential chronic risks. This study reveals potential health risks associated with wastewater irrigation due to heavy metal accumulation in edible vegetables, and therefore recommends further research on metal speciation, seasonal variation, and bioaccumulation at different crop growth stages.

## 1. Introduction

Wastewater reuse for irrigation has become an increasingly common practice worldwide. This approach is largely driven by growing water scarcity, rapid urbanization, and agricultural intensification [1,2,3]. Globally, more than 80% of wastewater is discharged untreated into the environment, and approximately 20 million hectares of land are irrigated with wastewater or wastewater-impacted water [4,5]. Although wastewater provides a reliable and nutrient-rich water source that supports urban and peri-urban agriculture [2,6]. However, it simultaneously acts as a vector for pollutants such as heavy metals, pharmaceuticals, and pathogens, facilitating their entry into the food chain [7]. Food contamination by pollutant represents a major global public health concern. According to the World Health Organization, nearly 600 million people—approximately one in ten worldwide—fall ill annually due to consumption of contaminated food. This results in around 420,000 deaths and 33 million Disability-Adjusted Life Years (DALYs) lost, with children under five accounting for approximately 40% of the burden [8,9].

Among pollutants, heavy metals are of particular concern in agricultural systems because they are non-biodegradable, persistent, and prone to bioaccumulate in soils, crops, and human tissues [10,11]. Some metals, such as Zn and Cu, are essential micronutrients required for enzymatic and physiological processes; however, others, including Cd and Pb, have no biological function and are toxic even at trace concentrations [11]. Chronic exposure to these metals has been associated with oxidative stress, neurotoxicity, renal dysfunction, carcinogenesis, and disruption of metabolic and endocrine pathways [12,13,14,15]. Mechanistically, heavy metals enter wastewater systems through industrial effluents, domestic sewage, urban runoff, and corroded plumbing systems, where they persist in soluble or particulate forms [11,16,17]. When such wastewater is reused for irrigation, these metals can adsorb onto soil colloids, undergo complexation with organic matter [7,17]. Subsequently become bioavailable to plants especially leafy vegetables through root uptake and translocation to edible tissues [18,19,20]. Once incorporated into crops, heavy metals can bio-magnify along the soil–plant–human continuum, posing chronic dietary exposure risks [21].

In Sub-Saharan Africa, wastewater irrigation has become an indispensable livelihood strategy for smallholder farmers, particularly in water-scarce urban and peri-urban areas during dry seasons [22,23]. However, the majority of wastewater used for irrigation in the region is partially treated or untreated, primarily due to aging infrastructure, inadequate funding, and weak regulatory enforcement [18]. Consequently, farmers and consumers are increasingly exposed to heavy metals through contaminated soil, crop uptake, and food consumption. Studies in Ghana, South Africa and Ethiopia, have reported elevated Cd, Pb, and Cr concentrations in wastewater-irrigated vegetables, often exceeding WHO permissible limits [10,23,24]. In Malawi has acknowledged the presence of heavy metals in wastewater and soils irrigated with effluent from local treatment plants [25,26]. For instance, Kraslawski et al. (2010) [25] investigated treatment efficiency and heavy metals at WWTPs in Blantyre, including Soche. The study reported that Cu, Ni, Cd, Cr and Pb concentration was below the detection limit during the wet season at Soche WWTP. Still, it did not assess metal accumulation in soil and crops. Malikula et al. (2022) [26] extended the scope by analyzing heavy metal and nutrient loads in wastewater, soil, and crops across multiple WWTPs, including Soche and Manase. However, their broader, multi-site approach concentrated on a limited range of crops, mainly exotic vegetables and emphasized seasonal variation and microbiological parameters, with less focus on comparative uptake between crop types or tissue parts.

In contrast, the present study offers a more detailed, site-specific investigation of the Soche WWTP and compares exotic and local vegetables irrigated with its effluent. It focuses on six commonly grown and consumed vegetables including three exotic Chinese cabbage (*Brassica rapa* subsp. *chinensis*), Mustard (*Brassica napus*), and Rape (*Brassica rapa*) and three indigenous; Pumpkin (*Cucurbita moschata*) leaves, Sweet potato (*Ipomoea batatas*) leaves, and Amaranthus (*Amaranthus retroflexus*) leaves. Moreover, little is known about heavy metal accumulation in both exotic and indigenous vegetables (including different plant parts like leaves and stems) irrigated with wastewater from the Soche WWTP. Hence, this study aimed at examining the heavy metals in wastewater, soil and both stems and leaves of vegetables irrigated with wastewater, and to assess potential health risks associated with consumption of these vegetables.

## 2. Materials and Methods

### 2.1. Study Area

This study was conducted in southern Malawi, Blantyre City, at the Soche WWTP (Figure 1). Soche WWTP is the oldest treatment plant in the country, which was built in the 1950s and extended in the 1970s to its current design flow of 4100 m^3^/day [27]. It is a conventional biological filter plant serving a population of over 40,000, the city’s main hospital and a few small industries such as car dealing, clothing making, and food processing. It is also the main tanker reception center for latrine and septic tank emptying [28]. Treated effluent from the plant is often used for irrigation by local farmers in nearby gardens. No inorganic or organic fertilizers were applied to the vegetable plots during cultivation.

### 2.2. Sample Collection

Sampling was conducted in November, providing a snapshot during the late dry/early rainy season. Seasonal variations in rainfall, irrigation frequency, and crop growth may influence metal accumulation, and future studies should include sampling across multiple seasons to capture temporal variability.

#### 2.2.1. Collection and Preservation of Wastewater Sample

An effluent sample was collected at the discharge outlet of the Soche WWTP using grab sampling, in triplicate (*n* = 3) [26,29]. Grab sampling was selected for its practicality in capturing a snapshot of effluent characteristics at a single point in time. Wastewater samples (1000 mL, in triplicate) were collected using acid-washed polyethylene bottles (United Scientific, Libertyville, IL, USA). To preserve the samples for metal analysis, 1 mL of 55% nitric acid (HNO_3_) (Glassworld & Chemical Suppliers (Pty) Ltd., Roodepoort, South Africa) was added immediately after collection [30]. Samples were stored in a cooler box and transported to the laboratory, where they were refrigerated at 4 °C and analyzed within 48 h to maintain sample integrity [30].

#### 2.2.2. Collection of Soil Sample

Fresh soil samples were collected from cultivated plots near the treatment plant where wastewater irrigation is practiced. A hand-held auger was used to collect samples from 5 to 30 cm depth [26,31,32]. The 5–30 cm depth was chosen because it represents the active root zone of vegetables and is the most affected by irrigation practices [33]. Visible debris and plant residues were removed, and the samples were placed in clean polyethylene bags (locally sourced) and transported to the laboratory for analysis.

#### 2.2.3. Collection of Vegetable Sample

Vegetable sampling followed the procedures described by Malikula et al. (2022) [26] and Abbas et al. (2023) [34]. Leaves and stems of the six vegetables were separately sampled to evaluate tissue-specific metal accumulation. The vegetables were collected in triplicate, yielding a total of 36 samples (6 species × 2 parts × 3 replicates). All samples were stored in clean, labeled polyethylene bags (locally sourced) and transported to the laboratory for further analysis.

### 2.3. Sample Preparation

#### 2.3.1. Wastewater Samples

Digestion of wastewater for metal analysis was carried out according to APHA standard method [30]. Three independent replicates of 50 mL of preserved wastewater were measured using a measuring cylinder (Poulten & Graf GmbH, Wertheim, Germany) and each transferred to a separate 100 mL volumetric flask (Poulten & Graf GmbH, Wertheim, Germany). The mixture of strong acids, typically a combination of 55% nitric acid (HNO_3_) (Glassworld & Chemical Suppliers (Pty) Ltd., Roodepoort, South Africa) and 31.5% hydrochloric acid (HCl) (Marlyn Chemicals Pty Ltd., Meyerton, South Africa) was added to the vessel. The acid mixture was used to dissolve metals from the wastewater matrix. The measured wastewater samples were then digested with a mixture of 10 mL concentrated HCl and 5 mL of concentrated HNO_3_. The mixtures were heated at 105 °C using a controlled hotplate (Daihan Scientific Ltd., Seoul, Republic of Korea) until digestion was completed. Distilled water (Lab Enterprises Ltd., Blantyre, Malawi) was added, and the sample was filtered with Whatman No. 42 filter paper (GE Healthcare, Chicago, IL, USA). The filtrate was topped up to 50 mL in a volumetric flask with distilled water, bringing it up to the mark. Blank was prepared by mixing 5 mL of concentrated HNO_3_ and 10 mL of concentrated HCl, and heated at the same temperature as the samples until digestion was completed [30].

#### 2.3.2. Soil Sample

Soil samples were prepared by first removing any visible debris or plant material. The samples were then oven-dried in a Griffin oven (Genlab Ltd., Widnes, UK) at 60 °C for 12 h. After drying, the soil was ground into a fine powder using a mortar and pestle and then sieved through a 2 mm mesh (Gilson Company Inc., Lewis Center, OH, USA) to homogenize the samples. One gram (1 g) of each well-dried and sieved soil sample was weighed in triplicate using an analytical balance (Adam Equipment, Oxford, UK). The prepared samples were then stored in clean beakers and sealed with parafilm to prevent contamination prior to digestion.

Sample digestion followed APHA standard method [30], where 1 g of the weighed of each soil sample was transferred to a digestion vessel, typically a teflon vessel (Savillex Corporation, Eden Prairie, MN, USA). The mixture of strong acids, typically a combination of 55% HNO_3_ (Glassworld & Chemical Suppliers (Pty) Ltd., Roodepoort, South Africa) and 70% perchloric acid (HClO_4_) (Glassworld & Chemical Suppliers (Pty) Ltd., Roodepoort, South Africa), was added to the vessel. The acid mixture facilitated the dissolution of metals from the soil matrix. The weighed soil sample was then digested with a mixture of 10 mL concentrated HClO_4_ and 5 mL of concentrated HNO_3_. The mixtures were heated at 105 °C using a controlled hotplate (Daihan Scientific Ltd., Seoul, Republic of Korea) until digestion was completed. Thereafter, distilled water (Lab Enterprises Ltd., Blantyre, Malawi) was added, and the sample was filtered with Whatman No. 42 filter paper (GE Healthcare, Chicago, IL, USA). The filtrate was topped up to 50 mL in a volumetric flask with distilled water, bringing it up to the mark. Blank was prepared by mixing 5 mL of concentrated HNO_3_ and 10 mL of concentrated HClO_4_ and heated at the same temperatures as samples until digestion.

#### 2.3.3. Vegetable Samples

The vegetable samples were washed using distilled water to remove debris. Then the leaves and stems were separated using a sterile knife. Thereafter, the sample was dried in Griffin Oven (Genlab Ltd., Widnes, UK) at a temperature of 60 °C for 12 h and ground into fine powder using a Kenwood blender (Kenwood Ltd., Hampshire, UK). Later, 1 g of each well-dried vegetable sample in triplicate was weighed using an analytical balance (Adam Equipment, Oxford, UK).

Sample digestion followed APHA standard method [30], where 1 g of each weighed vegetable sample was transferred to a digestion vessel, typically a teflon vessel. The mixture of strong acids, typically a combination of 55% HNO_3_ and 31.5% HCl, was added to the 100 mL vessel (Savillex Corporation, Eden Prairie, MN, USA). The acid mixture was used to break down the vegetable matrix and release the metals. The weighed vegetable samples were then digested with a mixture of 10 mL concentrated HCl (Marlyn Chemicals Pty Ltd., Meyerton, South Africa) and 5 mL of concentrated HNO_3_ (Glassworld & Chemical Suppliers (Pty) Ltd., Roodepoort, South Africa) [30]. The mixtures were heated at 105 °C using a controlled hotplate (Daihan Scientific Ltd., Seoul, Republic of Korea) until the digestion was completed. Distilled water was added, and the sample was filtered with Whatman No. 42 filter paper (GE Healthcare, Chicago, IL, USA). The filtrate was topped up to 50 mL in a volumetric flask with distilled water up to the mark. Blank was prepared by mixing 5 mL of concentrated HNO_3_ and 10 mL of concentrated HCl, and heated at the same temperature as the samples until digestion was completed.

### 2.4. Analytical Procedure

The analytical procedure followed the standard methods outlined by APHA standard method [30]. Heavy metal concentrations (Cd, Cr, Pb, Zn, and Cu) in wastewater, soil, and vegetable samples were determined using an Agilent 240FS Atomic Absorption Spectrophotometer (AAS) (Agilent Technologies, Santa Clara, CA, USA) equipped with a deuterium background corrector and element-specific hollow cathode lamps.

Instrument detection limits (mg/L) were: Cd 0.001, Cr 0.003, Pb 0.005, Zn 0.002, and Cu 0.002 (Appendix A). All analyses were conducted using a flame atomization system with an air-acetylene flame. Digested and filtered samples were transferred into clean, labeled 15 mL polyethylene test tubes, and exactly 10 mL of each sample was aspirated into the instrument using a nebulizer. Calibration was performed using external standards prepared by serial dilution of 1000 mg/L certified stock solutions (Inorganic Ventures, Christiansburg, VA, USA) in 2% nitric acid. Calibration curves exhibited good linearity with correlation coefficients (R^2^) exceeding 0.998. Blanks and a mid-level quality control standard were analyzed after every 10 samples to monitor accuracy and potential drift. Each sample was analyzed in triplicate, and the average value was reported. The final concentration was reported as mg/L for all samples.

### 2.5. Health Risk Assessment

The potential health risks associated with the consumption of vegetables contaminated with heavy metals were determined using the daily intake rate (DIR), target hazard quotient (THQ) and health risk index using (HRI) Equation (1), Equation (2) and Equation (3), respectively [24,26].(1)$$DIR=C_{\left(metal\;conc.\right)}\times C_{\left(factor\right)}\times D_{\left(veg.\;intake\right)}$$
where *DIR* signifies the daily intake rate, *C*_(*metal conc.*)_ denotes the heavy metal concentration in vegetables, *C*_(*factor*)_ represents the conversion factor (0.085), and *D*_(*veg.intake*)_ denotes the daily intake of vegetable [24].(2)$$THQ=\frac{E_{fr\;\;}\times ED\times DIR}{RFD\times BW\times AT}$$
where *THQ* signifies the target hazard quotient, *E_fr_* denotes the exposure frequency (365 days/year), *ED* represents the exposure duration (64 years), *DIR* signifies the daily intake rate, *RFD* denotes the reference oral dose (mg/kg/day), *BW* signifies the body weight of a consumer (60 kg), and *AT* denotes average time (*ED* × 365 days/year) [24,26].*HRI* = *∑THQ*(3)
where *HRI* signifies the health risk index, and *∑THQ* denoted the summation of all target hazard quotient [24].

### 2.6. Data Analysis

Quantitative data on heavy metal concentrations (Cd, Cr, Pb, Zn, and Cu) in wastewater, soil, and vegetable samples were first entered and cleaned in Microsoft Excel. Descriptive statistics (mean and standard deviation) were computed to summarize the data. One-way ANOVA was conducted in SPSS version 21 to test for significant differences in metal concentrations across sample types (wastewater, soil, vegetables) and within vegetable samples (leaves vs. stems) [10]. Statistical significance was set at *p* < 0.05. Where significant differences were detected, Duncan’s Multiple Range Test was used for post hoc comparisons to group samples based on their metal uptake. Principal Component Analysis (PCA) was performed to explore patterns of association among the five metals (Cd, Cr, Pb, Zn, and Cu) across wastewater, soils, and vegetable samples. PCA suitability was tested using KMO and Bartlett’s Test, and components with Eigenvalues > 1 were retained.

## 3. Results and Discussion

### 3.1. Concentration of Heavy Metals in Wastewater

In the current study, only Zn and Cu were detected in wastewater samples, while Cd, Cr, and Pb remained below detection limits (Table 1). This finding is consistent with earlier research [25,26] which similarly observed limited heavy metal presence in wastewater from Blantyre’s treatment plants, especially during the rainy season.

This study detected Zn at a concentration of 0.01 ± 0.0009 mg/L and Cu at 0.02 ± 0.0179 mg/L, while Cd, Cr, and Pb were all below detection limits (Table 1). These findings suggest low potential environmental risk based on current metal concentrations during sampling. The detection of Zn and Cu in the wastewater is primarily attributed to their widespread use in domestic and industrial applications. Zn commonly originates from sources such as galvanized pipes, roofing materials, and household detergents [35], while Cu is often released through plumbing corrosion, electrical wiring, and industrial effluents [36,37]. In particular, activities such as automotive repair, food processing, and textile manufacturing contribute to the presence of these metals. Therefore, occurrence in this study is consistent with inputs from domestic sewage, institutional discharges, and light industrial activities, as previously reported in Blantyre wastewater systems in Malawi [26,28].

The absence of Cd, Cr, and Pb may also be explained by hydrological dilution and physicochemical partitioning. Sampling occurred during the rainy season, when high stormwater inflows reduce contaminant concentrations through dilution and enhanced sedimentation. Furthermore, hydraulic overloading of the Soche WWTP (design capacity 4100 m^3^/day; [27]), likely promotes bypasses and resuspension of sediments, which can cause temporal variability in metal detection. Additionally, high-flow dynamics may affect metal distribution by either resuspending settled sediments or enhancing their settling prior to sample collection [38].

These findings contrast with earlier studies. For instance, Malikula et al. (2022) [26] reported significantly higher Cu levels (0.236 ± 0.17 mg/L), and Kraslawski et al. (2010) [25] detected trace amounts of Cd (0.002 ± 0.001 mg/L) at the same facility. The differences may be attributed to seasonal variation, differences in sampling protocols, and temporal shifts in the volume and type of industrial or domestic inputs. These discrepancies underscore the need for continuous multi-seasonal monitoring to accurately characterize heavy metal fluxes in wastewater.

Despite Zn and Cu concentrations being below the permissible limits set by MBS [39], their continuous discharge could lead to bioaccumulation and ecotoxicological impacts over time. Zn and Cu readily bind to organic matter and microbial biofilms, potentially altering enzymatic activity and nutrient cycling in receiving water bodies [40,41]. Conversely, the absence of Cr, Cd, and Pb should not be interpreted as an indication of low risk or permanent absence. These metals are persistent environmental toxicants and redox-sensitive, capable of remobilizing under fluctuating pH or redox conditions, as observed in other wastewater systems in Poland (Tytła and Widziewicz-Rzońca, 2023) [42]. Their non-detection in this study may therefore reflect temporary or seasonal variations rather than a true absence [10].

### 3.2. Concentrations of Heavy Metals in Soil

The analysis of soil irrigated with wastewater revealed detectable concentrations of all five targeted heavy metals, i.e., Zn, Cu, Cr, Pb, and Cd, with respective mean concentrations of Zn (56.40 ± 0.54 mg/kg), Cu (23.62 ± 0.55 mg/kg), Cr (38.14 ± 1.35 mg/kg), Pb (11.57 ± 0.06 mg/kg), and Cd (0.24 ± 0.01 mg/kg). As presented in Table 2, all metal concentrations except for zinc were within the permissible limits set by the WHO, indicating a relatively low environmental risk at the time of sampling. However, the elevated Zn concentration suggests long-term accumulation, likely due to continued exposure from municipal and industrial wastewater inputs, as soil tends to act as a sink for persistent contaminants [43].

The observed metal concentration trend (Cd < Pb < Cu < Cr < Zn) reflects both natural abundance and anthropogenic influence. For instance, Zn and Cu are commonly released from plumbing systems, industrial effluents, and detergents [26,28]. The presence of Cd and Pb, despite their non-detection in wastewater during this sampling, likely arises from historical accumulation, atmospheric deposition, and repeated irrigation cycles, as these metals have extended residence times and low leachability under neutral pH conditions [44].

The presence of these metals in soil poses various risks. Even at low concentrations, Cd and Pb are highly toxic due to their mobility and bioavailability, potentially affecting human health through crop uptake and food chain accumulation [27]. Elevated Zn can impair soil microbial communities and plant health, despite being an essential micronutrient [45].

### 3.3. Concentrations of Heavy Metals in Vegetables

Cr occurs mainly as trivalent Cr (III), an essential nutrient, and hexavalent Cr (VI), a toxic carcinogen [12]. Exposure to Cr (VI) via contaminated food can cause serious health issues such as liver and kidney damage, respiratory problems, and increased cancer risk [44]. In this study, total Cr concentration was detected in exotic and indigenous vegetables irrigated with wastewater, with 64% of samples exceeding the FAO/WHO safe limit of 2.3 mg/kg (Figure 2). ANOVA results showed significant differences in Cr concentrations among sample types (*p* < 0.05), and Duncan post hoc tests revealed distinct groupings that clarified the distribution of Cr across samples. Soil samples exhibited the highest Cr concentration (38.14 ± 1.35 mg/kg), forming a statistically distinct homogeneous group, far exceeding both wastewater and vegetable samples. In contrast, wastewater samples showed Cr concentrations below detection limits, placing them in the lowest statistical subset. This indicates that current wastewater inputs may not significantly contribute to Cr accumulation, and instead, soil acts as a long-term reservoir due to historical deposition, industrial discharge, and Cr’s strong affinity for clay particles and organic matter [45].

Highest Cr concentration was recorded in stems of *Brassica napus* (4.20 mg/kg) and *Cucurbita moschata* (4.65 mg/kg) which comprised a mid-high homogeneous subset, signaling their enhanced ability to absorb Cr from soil and translocate it to aerial parts. A recent study supports these observations: *Brassica napus* efficiently uptakes and mobilizes Cr via sulfate/phosphate transporters and allocates it to shoots, though its uptake can be moderated by amendments like silicon and selenium [46,47]. Similarly, *Cucurbita* species exhibit high Cr uptake attributed to their extensive root systems and high transpiration rates [48]. These findings are consistent with those of Malikula et al. (2022) [26], who also reported elevated Cr concentrations in vegetables grown at the same study site. However, they contrast with Asrade (2022) [49], who observed relatively lower Cr accumulation in leafy vegetables irrigated with wastewater in a different geographical setting. This discrepancy could be attributed to variations in wastewater composition, soil physicochemical properties, and local industrial discharge profiles, underscoring the spatial heterogeneity of heavy metals [50].

The low Cr concentrations in the stems and leaves of *Amaranthus retroflexus* where Cr was often below detection limits can be explained by the plant’s effective root-based exclusion mechanisms. A study on *Amaranthus retroflexus* showed that metals like Cr accumulate in roots, while stems and leaves remain low or undetectable [51]. Overall, Cr concentrations were higher in stems than in leaves due to more efficient xylem-mediated translocation and limited redistribution to leaf tissues [52]. Plants often sequester Cr in stem vacuoles or bind it to cell walls to prevent its movement into photosynthetically active leaves, thus minimizing oxidative damage [53].

Cd is a highly toxic metal with no biological role, posing serious health risks to humans, animals, and plants. Chronic exposure in humans can cause kidney damage, bone disorders, reproductive issues, and cancer [13,54]. In animals and crops, Cd accumulation impairs organ function and growth, leading to reduced productivity and health [55]. In this study, total Cd was detected in both exotic and indigenous vegetables irrigated with wastewater, with several samples exceeding the FAO/WHO maximum permissible limit of 0.2 mg/kg (Figure 3).

ANOVA revealed significant differences in Cd concentrations among sample types (*p* < 0.05), and Duncan post hoc tests identified distinct statistical groupings that clarified the distribution of Cd across wastewater, soil and vegetables.

Among all matrices, the soil sample exhibited detectable but moderate Cd concentrations (0.24 ± 0.01 mg/kg), forming a distinct statistical group. This indicates long-term accumulation driven by historical wastewater irrigation and Cd’s strong affinity for fine soil particles, organic matter, and clay minerals, which enable slow desorption into the rhizosphere [50,56]. Wastewater samples had Cd below detection limits, suggesting that current inputs are minor compared to legacy contamination. The highest Cd concentrations were recorded in the stems of *Brassica rapa* and *Cucurbita moschata*, which formed part of a mid-high statistical subset in the Duncan grouping. These species efficiently uptake Cd via calcium channels and metal transporters, with xylem loading facilitated by transpiration and vascular architecture [48,57]. In contrast, *Amaranthus retroflexus* leaves accumulated higher Cd than stems, likely due to active sequestration in leaf vacuoles mediated by phytochelatins, glutathione, and organic acids, while roots restrict translocation to aerial tissues [53,54,55].

Pb is a toxic metal with no biological benefit and acts as a potent neurotoxin, causing neurological, renal, reproductive, and developmental harm, especially in children and pregnant women [12,13,49]. In this study, Pb concentrations exceeded the FAO/WHO safe limit of 0.3 mg/kg across all vegetable species analyzed (Figure 4), underscoring widespread accumulation associated with wastewater-irrigated systems. ANOVA results revealed statistically significant differences in Pb concentrations across sample types (*p* < 0.05), and Duncan post hoc tests identified distinct homogeneous subsets, highlighting variation in tissue- and species-specific Pb accumulation. Soil samples exhibited elevated Pb levels (11.57 ± 0.06 mg/kg), forming a high statistical group, while wastewater samples recorded Pb below detection limits, suggesting that current irrigation practices may not be the immediate source. Instead, this points to legacy soil contamination, possibly from prior industrial emissions, atmospheric deposition, and vehicular exhausts, which are known sources of Pb persistence in agricultural soils [46].

Among vegetable samples, the highest Pb concentrations were recorded in the stems of *Brassica rapa* (4.09 ± 0.085 mg/kg), followed by other exotic species, while the lowest concentration was observed in *Brassica rapa* subsp. *chinensis*. These species were placed in mid-to-high statistical subsets, indicating a strong capacity for Pb uptake and stem localization. The preferential accumulation of Pb in stems rather than leaves is consistent with Pb’s low phloem mobility, and its strong affinity for cell wall components, lignin, and pectin, which promote sequestration in structural tissues [57,58]. Research by Duan et al. (2021) [59] and Sipos et al. (2023) [51] further supports that Pb, once absorbed by roots, is efficiently immobilized in xylem pathways or bound within parenchymal stem tissues, restricting its movement to leaf cells.

Lower Pb concentrations were consistently observed in leaf tissues, regardless of species, due to restricted xylem-to-phloem transfer and internal detoxification. Plants often limit Pb mobility to protect photosynthetically active leaf tissues, using vacuolar sequestration and apoplastic binding strategies [60]. Although leaves contained lower Pb concentrations than stems, their direct consumption—especially in raw or lightly cooked form—still poses a considerable dietary risk. This is particularly relevant for vegetables like *Brassica rapa* and *Amaranthus retroflexus*, which are commonly consumed for their leafy parts.

Cu is an essential micronutrient vital for both plant development and human health, supporting key physiological processes [54]. However, its benefits depend on concentration, as excessive levels can lead to toxicity and health complications [13]. In the present study, copper (Cu) was detected in all vegetable samples, with concentrations remaining within WHO permissible limits across exotic and indigenous species (Figure 5). ANOVA results showed significant differences among sample types (*p* < 0.05), and Duncan post hoc tests revealed clear groupings, with wastewater, soil, stems, and leaves forming distinct subsets.

Cu was also detected in wastewater and soil samples, suggesting that both environmental matrices are active contributors to Cu accumulation in vegetables. The presence of Cu in wastewater likely reflects discharges from domestic sources, small-scale industries, and runoff, which are known contributors to Cu loading in urban wastewater systems [26,28,61]. While the post hoc grouping placed wastewater in a lower statistical subset, its consistent detection indicates that ongoing irrigation inputs contribute to the Cu pool accessible to plants, especially with repeated application over time [61].

Soil samples formed the highest homogeneous subset in the post hoc analysis, indicating substantial Cu retention. This aligns with Cu’s strong binding affinity for organic matter, clay particles, and mineral surfaces, facilitating its long-term soil accumulation and reducing its immediate bioavailability [45,61]. Despite this, Cu was consistently detected in all vegetable samples, with significantly higher leaf concentrations than stems—particularly in indigenous species like *Amaranthus retroflexus*. This trend reflects Cu’s moderate mobility within plant systems and its preferential accumulation in metabolically active tissues, supporting critical functions such as photosynthesis, oxidative stress remediation, and enzyme activation [62].

Zn is an essential micronutrient crucial for human, animal, and plant health, supporting immune function, enzyme activity, and growth processes [13,63]. However, both deficiency and excess can negatively impact biological functions and productivity [63]. In this study, Zn was detected in all vegetable samples, with concentrations remaining below the FAO/WHO safety limit of 60 mg/kg, suggesting minimal health risk and highlighting the nutritional value of the vegetables (Figure 6). Post hoc analysis revealed significant differences across sample types, with soil forming the highest statistical group, indicating it as the main Zn reservoir. Wastewater, placed in the lowest group, had a limited direct contribution during sampling. Zn levels were generally higher in leaves than stems, and indigenous species like *Amaranthus retroflexus* and *Cucurbita moschata* showed greater accumulation, likely due to their physiological traits and higher transpiration rates [60]. These findings suggest that while soil is the primary source of Zn, plant-specific uptake mechanisms play a key role in Zn distribution. Although current levels pose no immediate risk, the presence of Zn in all samples and its tendency to build up in soil under prolonged wastewater irrigation highlight the need for monitoring to prevent future accumulation [61].

### 3.4. Principal Component Analysis

PCA results showed that the first principal component (PC1) had an Eigenvalue of 3.96, accounting for 79.2% of the total variance, while subsequent components explained less than 1 Eigenvalue each (Table 3. This indicates that the variability of all five heavy metals can be largely explained by a single underlying factor. Loadings of Zn (0.989), Cu (0.985), Cr (0.971), and Pb (0.923) were very strong on PC1, suggesting these metals share a common source, most likely wastewater effluent and historical soil contamination. By contrast, Cd (0.467) showed only a weak loading, suggesting that its distribution is more sporadic and may be influenced by different pathways such as legacy soil contamination or selective vegetable uptake (Table 4). The PCA results further confirm that heavy metals in wastewater-soil-vegetable systems are not independent but cluster under one main factor, pointing to common contamination pathways. This supports earlier findings that wastewater irrigation leads to simultaneous accumulation of multiple metals in edible plants, thereby amplifying health risks [33,64,65].

### 3.5. Health Risk Assessments

#### 3.5.1. Daily Intake Rate

The DIR of heavy metals refers to the number of heavy metals that an individual is exposed to or consumes daily [24]. DIR values of heavy metals were calculated to evaluate the health risks associated with vegetable consumption based on metal concentrations in vegetable tissues and typical ingestion rates, and compared against reference doses (RfDs) recommended by FAO/WHO (Table 5). The highest total DIRs were observed in *Brassica rapa* leaves (0.76 mg/kg/day), *Amaranthus retroflexus* leaves (0.86 mg/kg/day), and *Cucurbita moschata* stems (0.77 mg/kg/day), indicating significant bioaccumulation of multiple metals in these plant parts. This pattern is consistent with previous findings showing that leafy tissues accumulate more metals due to their higher metabolic activity and larger surface area exposed to contaminated irrigation water [66].

Among individual metals, Cr consistently had the highest DIR, followed closely by Cu, then Zn, with Cd and Pb showing the lowest overall DIR values. However, both Cd and Pb DIR exceeded safe thresholds in certain indigenous vegetables notably in the leaves of *Amaranthus retroflexus* and *Cucurbita moschata* and, to a lesser extent, in the stems of *Amaranthus retroflexus* and *Ipomoea batatas*. This suggests that while Cr, Cu, and Zn remain the dominant exposure risks, Cd and Pb present acute risks in specific edible tissues. A meta-analysis of *Amaranthus* species confirms that Cd accumulates preferentially in leaves, often exceeding stem levels due to species-specific uptake and translocation dynamics [67]. Similarly, field surveys across Bangladesh and Nigeria have documented that Cd and Pb exceed safe intake thresholds in leafy vegetables, even when soil and water levels are moderate [68]. The DIR ranking (Cr > Cu > Zn > Cd ≈ Pb) underscores a dual risk profile: chronic exposure to essential but potentially toxic elements (Cr, Cu, Zn) and acute exposure to highly toxic but less abundant elements (Cd, Pb) in specific species/tissues.

#### 3.5.2. Target Health Quotient and Health Risk Index

THQ is a ratio of the determined dose of a contaminant to the oral reference dose considered detrimental, whereas the HRI is the summation of all THQ values [24]. The THQ and HRI were used to assess the health risks associated with consuming vegetables irrigated with wastewater. A THQ or HRI value greater than 1 suggests a potential health concern. The results showed that Cr consistently posed the highest health risk among all analyzed metals, followed by Pb and Cu (Table 6). The highest total HRI was recorded in the stem of *Brassica rapa* (HRI = 92.28), followed by the leaf of *Brassica rapa* subsp. *chinensis* (HRI = 82.24), and the stem of *Brassica napus* (HRI = 75.04). These values were driven primarily by extremely high THQs for Cr and Pb, indicating significant health risks associated with consuming these vegetables, particularly their stems. While exotic vegetables generally exhibited higher THQ values for Cr and Pb, the indigenous vegetables showed elevated THQs for Cu and, to a lesser extent, Zn. For instance, the leaf of *Amaranthus retroflexus* recorded a Cu THQ of 4.87 and Zn THQ of 2.05, resulting in a total HRI of 31.57. Similarly, *Cucurbita moschata* leaf showed Cu THQ of 4.47 and Zn THQ of 1.48, contributing to a high HRI of 50.04. These findings highlight the significant role Cu and Zn play in driving health risks among indigenous vegetables. The results also showed that stems had higher HRI values than leaves in four out of six vegetable species, including *Brassica rapa*, *Brassica napus*, *Cucurbita moschata*, and *Ipomoea batatas*. However, in *Brassica rapa* subsp. *chinensis* and *Amaranthus retroflexus*, the leaves showed higher HRIs, suggesting that metal accumulation patterns are both species- and tissue-specific.

Several factors may explain these findings. Firstly, Cr and Pb are relatively immobile in vegetable samples, often accumulating in stems due to strong binding with cell walls and limited translocation via the phloem [58]. This may account for the higher THQs for Cr and Pb observed in the stems of *Brassica* species. On the other hand, Cu and Zn are essential micronutrients, actively absorbed and translocated by plants. Their mobility, combined with greater metabolic activity and transpiration in leaves, facilitates higher accumulation in leaf tissues [61]. Moreover, indigenous species like *Amaranthus retroflexus* and *Cucurbita moschata* are known for their efficient uptake and tolerance to micronutrient stress, which may contribute to elevated metal concentrations in their edible parts [55]. Foliar exposure is another possible pathway; leafy vegetables irrigated with wastewater are also exposed to atmospheric deposition and splash contamination, which can enhance surface accumulation of metals like Zn and Cu [66]. Environmental factors such as soil pH, organic matter, and redox conditions further influence metal bioavailability and uptake. For instance, acidic or slightly acidic soils common in agricultural zones enhance the solubility and mobility of Cu and Zn, promoting their uptake by plants [18].

## 4. Conclusions

This study assessed the concentrations of heavy metals (Cd, Cr, Pb, Zn, and Cu) in wastewater, agricultural soils, and commonly consumed vegetables grown around the Soche WWTP in Malawi. While wastewater and soils generally showed metal concentrations within permissible limits except for elevated Zn in soils which reached 56.4 ± 0.5 mg/kg, exceeding the WHO permissible limit of 36 mg/kg. This elevated concentration was likely due to prolonged wastewater irrigation. Vegetables exhibited species- and tissue-specific accumulation patterns, with certain leafy vegetables such as *Brassica rapa*, *Brassica rapa* subsp. *chinensis*, and *Cucurbita moschata* showing the highest accumulation of Cr, Cd, and Pb, whereas *Amaranthus retroflexus* exhibited the lowest accumulation. Although Zn and Cu are essential micronutrients for plant growth and human health, their accumulation beyond recommended limits may still pose long-term risks, especially when combined with toxic metals such as Cd and Pb. It is important to note that this study quantified total elemental concentrations using Atomic Absorption Spectrophotometry (AAS), which does not differentiate between chemical forms. Since toxicity depends strongly on chemical speciation (e.g., Cr^3+^ being less toxic than Cr^6+^), future research should incorporate speciation analysis. Overall, the findings of this study underscore the importance of integrating environmental monitoring, food safety, and public health risk assessments. To mitigate these risks, we recommend the following: farmers should adopt vegetables with lower metal accumulation; health and agricultural authorities should implement regular monitoring of wastewater, soils, and crops. Therefore, this study recommends further research on heavy metal speciation, seasonal variations, and bioaccumulation at different crop growth stages.

## Figures and Tables

**Figure 1 ijerph-22-01614-f001:**
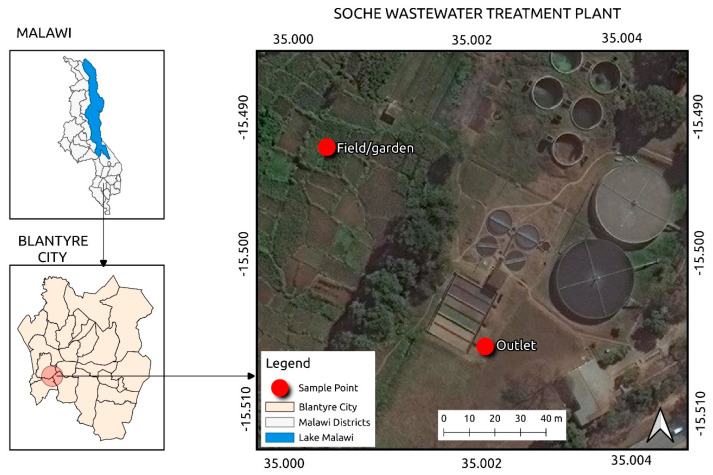
Location of the study area.

**Figure 2 ijerph-22-01614-f002:**
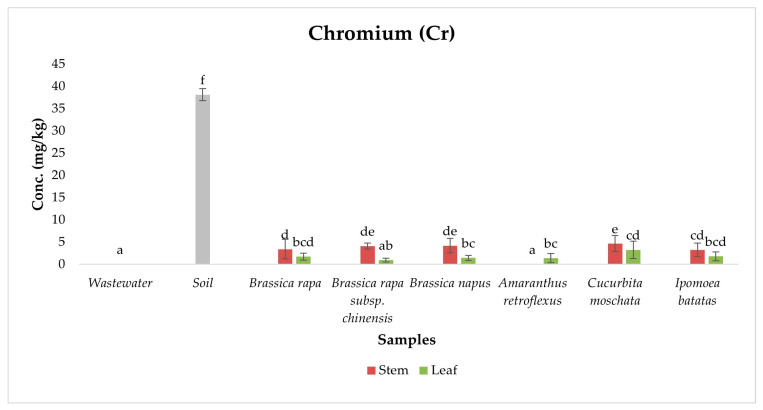
Chromium concentration in samples. Different lowercase letters above the bars indicate statistically significant differences among sample types (*p* < 0.05, Duncan’s Multiple Range Test).

**Figure 3 ijerph-22-01614-f003:**
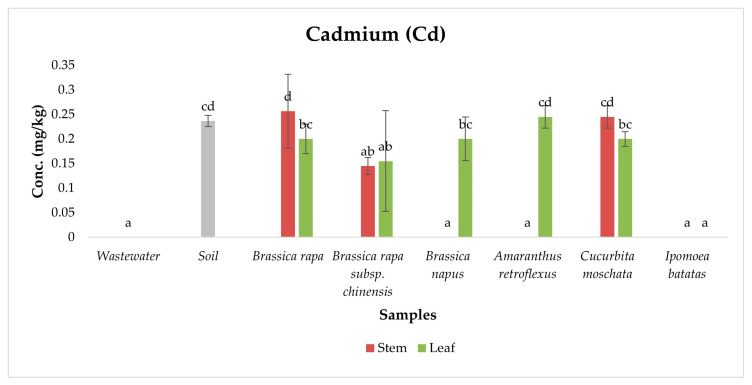
Concentration of Cadmium in samples. Different lowercase letters above the bars indicate statistically significant differences among sample types (*p* < 0.05, Duncan’s Multiple Range Test).

**Figure 4 ijerph-22-01614-f004:**
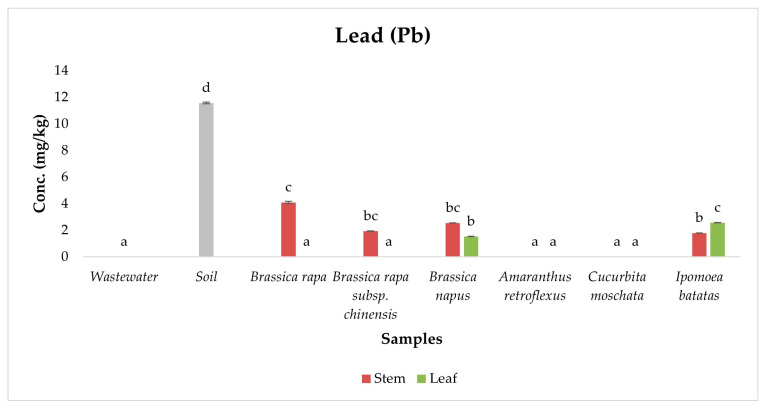
Concentration of Lead in samples. Different lowercase letters above the bars indicate statistically significant differences among sample types (*p* < 0.05, Duncan’s Multiple Range Test).

**Figure 5 ijerph-22-01614-f005:**
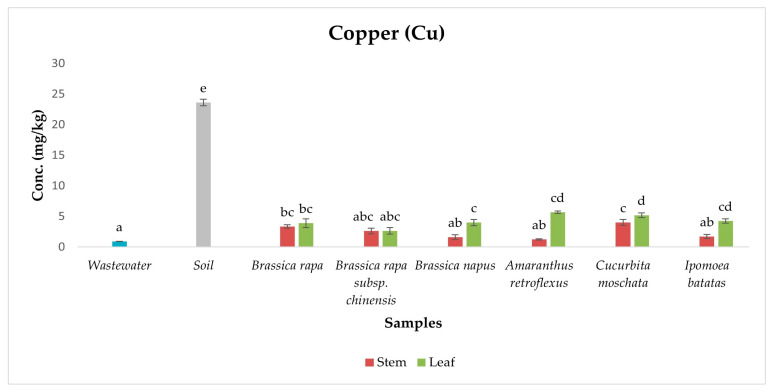
Concentration of Copper in samples. Different lowercase letters above the bars indicate statistically significant differences among sample types (*p* < 0.05, Duncan’s Multiple Range Test).

**Figure 6 ijerph-22-01614-f006:**
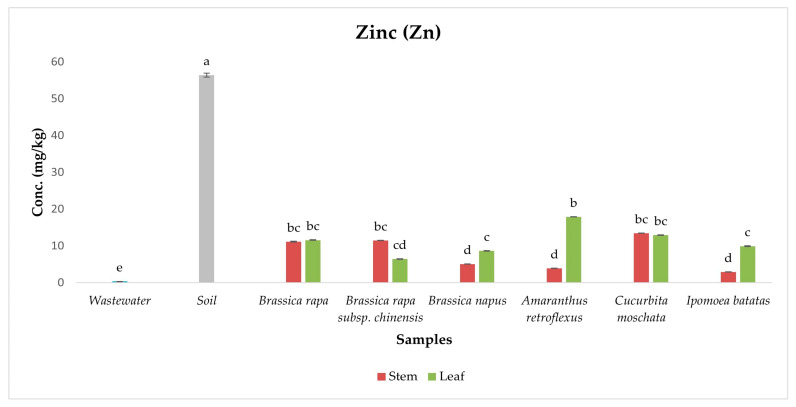
Concentration of Zinc in samples. Different lowercase letters above the bars indicate statistically significant differences among sample types (*p* < 0.05, Duncan’s Multiple Range Test).

**Table 1 ijerph-22-01614-t001:** Heavy metal concentrations in effluent from Soche WWTP.

Heavy Metals	Cd (mg/L)	Cr (mg/L)	Pb (mg/L)	Zn (mg/L)	Cu (mg/L)
Soche	BDL	BDL	BDL	0.01 ± 0.0009	0.02 ± 0.0179
MBS	0.05	0.05	0.05	5	2
WHO Guidelines	0.05	0.003	0.01	0.2	2

BDL: Below Detection Limit.

**Table 2 ijerph-22-01614-t002:** Heavy metal concentrations in soil from fields of Soche WWTP.

Heavy Metals	Cd (mg/kg)	Cr (mg/kg)	Pb (mg/kg)	Zn (mg/kg)	Cu (mg/kg)
Soche	0.24 ± 0.013	38.14 ± 1.351	11.57 ± 0.063	56.40 ± 0.543	23.62 ± 0.554
WHO	0.8	100	85	36	50

**Table 3 ijerph-22-01614-t003:** Total variance explained by principal component analysis of heavy metal concentrations in wastewater, soil, and vegetable samples.

Component	Initial Eigenvalues			Extraction Sums of Squared Loadings		
	Total	% of Variance	Cumulative %	Total	% of Variance	Cumulative %
1	3.960	79.202	79.202	3.960	79.202	79.202
2	0.859	17.170	96.372			
3	0.140	2.801	99.174			
4	0.032	0.641	99.814			
5	0.009	0.186	100.000			

**Table 4 ijerph-22-01614-t004:** Component loadings of heavy metals on the first principal component (PC1) extracted using Principal Component Analysis.

	Component
	1
Zinc	0.989
Copper	0.985
Chromium	0.971
Lead	0.923
Cadmium	0.467

**Table 5 ijerph-22-01614-t005:** Daily Intake Rate of Vegetables.

Name of Vegetable	Portion of Vegetable	Daily Intake Rate (mg/kg/day)					
		Cr	Cd	Pb	Zn	Cu	Total *DIR*
*Amaranthus retroflexus*	Stem	0	0	0	0.13	0.04	0.17
	Leaf	0.05	0.01	0	0.61	0.19	0.86
*Brassica napus*	Stem	0.14	0	0.09	0.17	0.06	0.46
	Leaf	0.05	0.01	0.05	0.30	0.14	0.55
*Brassica rapa*	Stem	0.12	0.01	0.14	0.38	0.11	0.76
	Leaf	0.06	0.01	0	0.40	0.13	0.6
*Brassica rapa* subsp. *chinensis*	Stem	0.14	0	0.07	0.39	0.09	0.69
	Leaf	0.22	0.01	0	0.22	0.09	0.54
*Cucurbita moschata*	Stem	0.16	0.01	0	0.46	0.14	0.77
	Leaf	0.11	0.01	0	0.44	0.18	0.74
*Ipomea batatas*	Stem	0.11	0	0.06	0.10	0.06	0.33
	Leaf	0.06	0	0.09	0.34	0.15	0.64
RfD_0_ (mg/kg/day) (FAO/WHO)		0.003	0.001	0.0035	0.3	0.04	

**Table 6 ijerph-22-01614-t006:** Target Health Quotient and Health Risk Index.

Name of Vegetable	Portion of Vegetable	Daily Intake Rate (mg/kg/day)					
		Cr	Cd	Pb	Zn	Cu	*HRI* = *∑THQ*
*Amaranthus retroflexus*	Stem	0	0	0	0.44	1.06	1.50
	Leaf	16.23	8.42	0	2.05	4.87	31.57
*Brassica napus*	Stem	48.11	0	24.98	0.58	1.38	75.04
	Leaf	16.63	6.88	14.98	0.99	3.44	42.92
*Brassica rapa*	Stem	39.15	8.82	40.17	1.28	2.86	92.28
	Leaf	19.75	6.88	0	1.33	3.34	31.29
*Brassica rapa* subsp. *chinensis*	Stem	46.10	4.98	18.89	1.31	2.25	74.43
	Leaf	73.91	5.33	0	0.74	2.26	82.24
*Cucurbita moschata*	Stem	53.28	8.42	0	1.54	3.44	66.68
	Leaf	37.22	6.875	0	1.48	4.47	50.04
*Ipomea batatas*	Stem	37.26	0	17.61	0.33	1.46	56.66
	Leaf	20.57	0	25.14	1.13	3.64	50.48

## Data Availability

The data presented in this study are available from the corresponding. authors upon reasonable request.

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
