# Peer review of "Assessment of Heavy Metal Accumulation in Wastewater–Receiving Soil–Exotic and Indigenous Vegetable Systems and Its Potential Health Risks: A Case Study from Blantyre, Malawi"

_ijerph, 2025, doi:10.3390/ijerph22111614_

Round 1

Reviewer 1 Report

Comments and Suggestions for Authors

I have some points about the article entitled “Assessment of Heavy Metal Accumulation in Wastewater–Receiving Soil–Exotic and Indigenous Vegetable Systems and Their Potential Health Risks: A Case Study from Blantyre, Malawi” submitted in the International Journal of Environmental Science and Public Health. The manuscript is well written however it need some substanitial improvement to meet the quality of journal and is suitable for publication after major revision.

General suggestions

Proper arrangement of the references cited in the manuscript according to the Journal’s Guidelines for the Authors. Also references are old.

Most importantly:

  • Introduction section is small with long sentences.
  • Methodology section is long without proper citations please make it concise with proper citation and use just important information.
  • Looks that whole the analytical procedure was performed without reference material if added what was the range? how to check the analytical authenticity?

Abstract

Abstract always stands alone therefore pay attention on it.

  1. Where is specific gap in knowledge or problem being addressed?
  2. Lines 25-37: please add the concentration value of the metals.
  3. Add the reference allowable limit where required.
  4. Methodology section is too long please making it concise and revise the abstract with meaningful conclusion.

Introduction

Introduction of the paper is too much important, a good introduction must cover all the aspects of the studied problem. The current introduction of the article needs some of improvement to meet the quality standard of the journal.

  1. Sentences are too long with single reference.
  2. The structure of the introduction is not proper Important background studies relevant to the topic are missing; please include recent literature (last 5 years) to strengthen context.
  3. Line 53: Which public health issue these metals exert to public of Malawi?
  4. Line 59: leafy vegetables only single reference.
  5. Only single study for references in Malawi, please add some more references to strengthen your objective.

Materials and Methods 

  1. Line 98: Collection and Preservation of Wastewater Sample.  which study you followed?
  2. Line 109: how many replicates?
  3. Line 115: Collection of Vegetable Sample, which study you followed?
  4. Line 121: Wastewater Samples: have you developed your own methods? Wonder how all the study was performed without a standard protocol?
  5. Same problem with section 2.3.2 soil samples and 2.3.3 vegetable samples.
  6. Section 2.3.3: no blank samples.
  7. Lines 122-123: how many replicates?
  8. Section 2.4, line 182: (AAS) what was the detection limit of the instrument?
  9. Line 197: no need of Table 1 in the main body. This is not your main study you can keep it as a supplementary data.
  10. Section 2.6, line 199: how this equation was generated? Without any citation
  11. Line 224: what was the detection limit?
  12. Figures 2,3,4,5,6: what does a,b,c represents, no detail available?

Conclusions

Line 513: What was the permissible limit and what was the value of Z?

Author Response

ABSTRACT

Comments 1: Where is specific gap in knowledge or problem being addressed?.

Response 1: Thank you for pointing this out. In the revised abstract we now clearly state the specific knowledge gap: limited data on heavy-metal accumulation in both exotic and indigenous vegetables irrigated with wastewater in Malawi, particularly around the Soche Wastewater Treatment Plant.

Comments 2: Lines 25-37: please add the concentration value of the metals.

Response 2: Thank you for pointing this out. We have now inserted representative concentration values for heavy metals in wastewater, soil, and vegetables directly into the abstract.

Comments 3: Add the reference allowable limit where required.

Response 3: Thank you for the suggestion. In the revised abstract, we have included FAO/WHO and Malawi Bureau of Standards (MBS) permissible limits alongside the measured concentrations.                                    

Comments 4: Methodology section is too long please making it concise and revise the abstract with meaningful conclusion.

Response 4: Thank you for pointing this out. We have revised the methodology section, by removing surplus information such as Table 1 is provided as a supplementary data.

INTRODUCTION

Comments 1: Sentences are too long with single reference.

Response 1: Thank you for pointing this out. We have added references and long sentences have been shortened.

Comments 2: The structure of the introduction is not proper Important background studies relevant to the topic are missing; please include recent literature (last 5 years) to strengthen context.

Response 2: Thank you for pointing this out. We have incorporated recent literature from 2020-2025 [31, 58,59, 1, 60, 61, 25].

Comments 3: Line 53: Which public health issue these metals exert to public of Malawi?

Response 3: Thank you for pointing this out. We have added a sentence to include some of the public health issues these metals can exert such neurological, respiratory, urinary and cardiovascular disorders due to immune-modulation, oxidative and inflammatory mechanisms.

Comments 4: Line 59: leafy vegetables only single reference.

Response 4: Thank you for pointing this out. We have added 2 more references for leafy vegetables (61,62).

Comments 5:    Only single study for references in Malawi, please add some more references to strengthen your objective.

Response 5: Thank you for the suggestion. We have added some more references for Malawi which include [18,22,63,64].

METHODOLOGY

Comments 1: Line 98: Collection and Preservation of Wastewater Sample.  which study you followed?

Response 1: Thank you for pointing this out. We have cited the studies that we followed during wastewater sample collection and preservation [22,57].

Comments 2: Line 109: how many replicates?

Response 2: Thank you for pointing this out. We have indicated 3 replicates.

Comments 3: Line 115: Collection of Vegetable Sample, which study you followed?

Response 3: Thank you for pointing this out. We have cited the studies that we followed during collection of vegetable samples [22,54].

Comments 4: Line 121: Wastewater Samples: have you developed your own methods? Wonder how all the study was performed without a standard protocol?

Response 4: Thank you for the suggestion. We have indicated the standard protocol adopted from APHA (2017).

Comments 5: Same problem with section 2.3.2 soil samples and 2.3.3 vegetable samples.

Response 5: Thank you for pointing this out. We have indicated the standard protocol adopted from APHA (2017).

Comments 6: Section 2.3.3: no blank samples.

Response 6: Thank you for pointing this out. We have included the bank sample in section 2.3.3.

Comments 7: Lines 122-123: how many replicates?

Response 7: Thank you for pointing this out. We have indicated 3 replicates in line 122-123.

Comments 8: Section 2.4, line 182: (AAS) what was the detection limit of the instrument?

Response 8: Thank you for pointing this out. We have indicated the instrumental detection limit in line 215.

Comments 9: Line 197: no need of Table 1 in the main body. This is not your main study you can keep it as a supplementary data.

Response 9: Thank you for pointing this out. We have moved Table 1 to the supplementary data.

Comments 10: 10.         Section 2.6, line 199: how this equation was generated? Without any citation.

Response 10: Thank you for pointing this out. The citations for the all the equations used are in line 234.

Comments 11: Line 224: what was the detection limit?

Response 11: Thank you for pointing this out. We have indicated the detection limits in line 221.

Comments 12: Figures 2,3,4,5,6: what does a,b,c represents, no detail available?

Response 12: Thank you for pointing this out. Different lowercase letters (a–c) above the bars indicate statistically significant differences (p < 0.05) among sample types according to Duncan’s Multiple Range Test.

CONCLUSION

Comments 1: Line 513: What was the permissible limit and what was the value of Z?

Response 1: Thank you for pointing this out. We have included the permissible limit and the Z value.

Reviewer 2 Report

Comments and Suggestions for Authors

1.About the scientific expressions. (1)There are minor inconsistencies in terminology and formatting(e.g.the manuscript inconsistently uses both "vegetable sample" and "plant tissue". It will be better if the names are unified). (2) You incorrectly wrote Perchloric acid to HclO4, please check all the expressions of chemical reagents. (3)Please unify and clarify the units (in the tables and in text) to make your presentation clearer.

2.About the graphs.In Figures 2 to 6, the wastewater and soil bars are all grouped above Brassica rapa. Please separate them so the graph will be clearer and not confusing.

3.About the conclusion and the contents. (1) You analyzed zinc and copper concentrations, but since both are essential elements for plants, this alone cannot indicate any problem. It would be better to compare your results with data from previous studies and include more international safety standards as well as detailed analyses. (2) The study measured elemental concentrations using AAS, which detects total element content but not specific chemical forms. If you add speciation analysis(e.g. Distinguishing chromium species(Cr3+and Cr6+) or forms of the original compounds) would provide a clearer picture of potential toxicity(You can use pH to make a simple analysis).

4. About statistic and method -- You can add more details to better illustrate your ideas. It is suggested to provide details on statistical methods (e.g., p-values for ANOVA, as referenced on page 14 for HRI comparisons) and AAS validation (e.g., calibration curves, detection limits, page 1).

Comments on the Quality of English Language

1. Line 224 "In the current study, only Zn and Cu were detected in wastewater samples, while

Cd, chromium Cr, and Pb remained below detection limits. " Delete the"chromium" before Cr.

2. Please review all sentences, remove surplus expressions, and correct grammatical mistakes.

Author Response

Comments 1: About the scientific expressions. (1)There are minor inconsistencies in terminology and formatting(e.g.the manuscript inconsistently uses both "vegetable sample" and "plant tissue". It will be better if the names are unified). (2) You incorrectly wrote Perchloric acid to HclO4, please check all the expressions of chemical reagents. (3)Please unify and clarify the units (in the tables and in text) to make your presentation clearer.

Response 1: Thank you for pointing this out. We have used vegetable sample instead of plant tissue throughout the manuscript. Hcl04 has been correctly written to HClO4 and all expressions of chemical reagents have been checked. The units in the text and tables have been unified throughout the whole document.

Comments 2: About the graphs. In Figures 2 to 6, the wastewater and soil bars are all grouped above Brassica rapa. Please separate them so the graph will be clearer and not confusing.

Response 2: We thank the reviewer for this suggestion regarding the graphs. We have separated wastewater, and soil bars in all the graphs.  

Comments 3: About the conclusion and the contents. (1) You analyzed zinc and copper concentrations, but since both are essential elements for plants, this alone cannot indicate any problem. It would be better to compare your results with data from previous studies and include more international safety standards as well as detailed analyses. (2) The study measured elemental concentrations using AAS, which detects total element content but not specific chemical forms. If you add speciation analysis (e.g. Distinguishing chromium species (Cr3+and Cr6+) or forms of the original compounds) would provide a clearer picture of potential toxicity (You can use pH to make a simple analysis).

Response 3: Thank you for pointing this out. We have revised the conclusion to clarify the dual role of Zn and Cu as both essential micronutrients and potential toxicants when present above recommended levels. As correctly pointed out, our study employed Atomic Absorption Spectrophotometry (AAS), which quantifies total elemental concentrations but does not differentiate between chemical species. We have revised the manuscript conclusion to explicitly acknowledge this limitation. We now emphasize that while our findings provide a clear indication of heavy metal contamination, the actual toxicity depends strongly on the chemical form (e.g., Cr³⁺ being relatively less toxic compared to Cr⁶⁺). We have also recommended that future studies on wastewater-irrigated soils and vegetables in Malawi should incorporate speciation analysis.

Comments 4: About statistic and method -- You can add more details to better illustrate your ideas. It is suggested to provide details on statistical methods (e.g., p-values for ANOVA, as referenced on page 14 for HRI comparisons) and AAS validation (e.g., calibration curves, detection limits, page 1).

Response 4: Thank you for pointing this out. In response, we have expanded the description of our statistical analyses in the Methods section. Regarding AAS validation, all relevant details previously in Table 1, including calibration curves, detection limits, recovery tests, and replicate analyses, have now been moved to the Supplementary Data (Table S1) for clarity.

Comments 5: Line 224 "In the current study, only Zn and Cu were detected in wastewater samples, while Cd, chromium Cr, and Pb remained below detection limits. " Delete the"chromium" before Cr.

Response 5: Thank you for pointing this out. We have deleted Chromium before Cr.

Comments 6: Please review all sentences, remove surplus expressions, and correct grammatical mistakes.

Response 6: Thank you for pointing this out. We have reviewed all sentences and removed surplus expression in the introduction and methodology. Grammatical mistakes have been checked.

Reviewer 3 Report

Comments and Suggestions for Authors

I have carefully reviewed the manuscript entitled “Assessment of Heavy Metal Accumulation in Wastewater–Receiving Soil–Exotic and Indigenous Vegetable Systems and Their Potential Health Risks: A Case Study from Blantyre, Malawi” submitted by Chiutula et al. to a MDPI journal International Journal of Environmental Research and Public Health. This study determined concentrations of cadmium (Cd), chromium (Cr), lead (Pb), zinc (Zn), and copper (Cu) in wastewater, soils, and six different vegetables including three exotic and three indigenous cultivated near the Soche Wastewater Treatment Plant in Blantyre. Finally, this study reveals potential health risks associated with wastewater irrigation due to heavy metal accumulation in edible vegetables, and therefore recommends further research on metal speciation, seasonal variation, and bioaccumulation at different crop growth stages.

The experimental design, statistical methods, and testing and calculation procedures were correct. This study provided some useful data and these data could advance the understanding of healthy vegetable cultivation in local. Although this study is very simple and the data volume is limited, I still recommend accepting this manuscript for publication after some revisions, considering actual production guide significance. Some comments are as the following:

  1. How many repetitions were made when collecting samples? In other words, what is the sample size?
  2. A total of six vegetables were used here. Which vegetable has the strongest ability to accumulate heavy metals, and which one has the lowest? What is the guiding value of these information for local vegetable production and planting arrangements.
  3. Supplementing the information on fertilization during vegetable cultivation.
  4. It should supplement some discuss associated with the long-term effect of wastewater irrigation on vegetable production, and also, propose some feasible methods to decline the negative effect of wastewater irrigation on vegetable production in local.

Author Response

Comments 1: How many repetitions were made when collecting samples? In other words, what is the sample size?

Response 1: Thank you for pointing this out. We have clarified the sample size and replication in the Materials and Methods. Three independent replicates were collected for wastewater, soils, and each vegetable species.

Comments 2: A total of six vegetables were used here. Which vegetable has the strongest ability to accumulate heavy metals, and which one has the lowest? What is the guiding value of these information for local vegetable production and planting arrangements.

Response 2: We thank the reviewer for this insightful comment. In the revised manuscript, we have incorporated species-specific heavy metal accumulation patterns and the guiding value in the conclusion section.

-Comments 3: Supplementing the information on fertilization during vegetable cultivation.

Response 3: Thank you for pointing this out. We have clarified in section 2.1 that no inorganic or organic fertilizers were applied during cultivation, and effluent served as both the water and nutrient source.

Comments 4: It should supplement some discuss associated with the long-term effect of wastewater irrigation on vegetable production, and also, propose some feasible methods to decline the negative effect of wastewater irrigation on vegetable production in local.

Response 4: Thank you for pointing this out. We agree with this comment. We have proposed feasible methods to decline the negative effects of wastewater in the conclusion section.

Reviewer 4 Report

Comments and Suggestions for Authors

The comments are attached.

Author Response

Comments 1: Please write some details about sampling procedure while considering variables like season etc.

Response 1: Thank you for pointing this out. In the revised manuscript, we have included details about the timing and seasonal context of sampling in the Sample Collection subsection 2.2 of the Methods section. Specifically, we note that sampling was conducted in November, providing a snapshot during the late dry/early rainy season. We also acknowledge that seasonal variations in rainfall, irrigation frequency, and crop growth may influence heavy metal accumulation, and recommend that future studies include sampling across multiple seasons to capture temporal variability.

Comments 2: Advanced multivariate tools such as Principal Component Analysis (PCA) or heatmaps should be employed to better illustrate relationships among parameters and to support stronger conclusions.

Response 2: We thank the reviewer for this suggestion regarding Principal Component Analysis. We have added PCA in section 3.4.

Comment 3: The authors should focus on improving discussion with mechanistic insights with international studies.

Response 3: Thank you for pointing this out. We have improved the discussion by providing some mechanistic insights and some international standards.

Comments 4: In conclusion, please include some actionable recommendations for the stakeholders like farmers, health concern authorities.

Response 4: Thank you for pointing this out. We have included actionable recommendations in the conclusion. See line 609-612

Round 2

Reviewer 1 Report

Comments and Suggestions for Authors

Although the authors showed alot of improvement but in some points there is no proper respones.

  1. Introduction section need further improvement only 2-3 lines or not enough to strenghten the manuscript or to meet the publiction criteria of the journal
  2.  In most of the comments line number is not mentioned how to find your answer. Be professional please.
  3. How about the refreences material seems that no refrence material was inculded so, how to confirm the authenticity of whole procedure ?
  4. Comments 12: Figures 2,3,4,5,6: what does a,b,c represents, no detail available?

( Different lowercase letters (a–c) above the bars indicate statistically significant differences (p < 0.05) among sample types according to Duncan’s Multiple Range Test).

This statement should be included below each figure not here

Author Response

ROUND 2: RESPSONSE TO REVIEWER 1

Comments 1: Introduction section need further improvement only 2-3 lines or not enough to strengthen the manuscript or to meet the publication criteria of the journal

Response 1: Thank you for pointing this out. The introduction has been substantially revised to enhance its scientific depth, logical flow, and contextual linkage from global to regional (Sub-Saharan Africa) and local (Malawi) perspectives. Specifically, we have expanded the discussion on the global drivers of wastewater reuse, associated health risks, and heavy metal dynamics in wastewater-irrigated systems. We have also added supporting data and references from FAO, WHO, and UN-Water reports to strengthen the background. See section 1, page 2, line 68-104.

Comments 2: How about the reference’s material seems that no reference material was included so, how to confirm the authenticity of whole procedure?

Response 2: Thank you for pointing this out. The methodology employed in this study follows standardized and widely recognized analytical protocol, all of which have been appropriately cited within the Methods section to ensure authenticity and reproducibility. Specifically,

-           Sample collection and preservation procedures for wastewater, soil, and vegetables are referenced to Malikula et al. (2022), Abbas et al. (2023), and APHA (2017) (See page 4, section 2.2.1 line 146, 151, 153, section 2.2.2, page 4, line 157, 158, Section 2.2.3 page 5, line 162-163.).

-          Sample digestion methods for wastewater, soil, and vegetables explicitly follow APHA (2017) standards (see Lines 168–214), which are globally accepted reference methods for environmental and water quality analysis. See section 2.31, page 5, line 171, 186, section 2.3.2, page 5 line 196, section 3.3.3, page 6 line 218, 225.

-          Analytical determination of metals was conducted using an Agilent 240FS Atomic Absorption Spectrophotometer (AAS), following the American Public Health Association (APHA, 2017) procedure. See section 2.4, page 6, line 234.

-          Health Risk Assessment equations and assumptions are adapted from peer-reviewed literature, specifically Malikula et al. (2022) and Makanjuola et al. (2019) (see section 2.6, page 6 line 254).

Comments 3: Comments 12: Figures 2,3,4,5,6: what does a,b,c represents, no detail available? (Different lowercase letters (a–c) above the bars indicate statistically significant differences (p < 0.05) among sample types according to Duncan’s Multiple Range Test). This statement should be included below each figure not here

Response 3: Thank you for pointing this out. We have added the statement below the figures throughout the whole manuscript. See line 361-362, line 396-398, line 426-427, line 455-456, line 490-491.

Comments 4: In most of the comments line number is not mentioned how to find your answer. Be professional please.

Response 4: Thank you for pointing this out. To Address this comment, we have indicated the lines of the comments that was provided during round 1 which are below.
